# An Investigation into the Effects of a Curcumin Extract (Curcugen^®^) on Osteoarthritis Pain of the Knee: A Randomised, Double-Blind, Placebo-Controlled Study

**DOI:** 10.3390/nu14010041

**Published:** 2021-12-23

**Authors:** Adrian L. Lopresti, Stephen J. Smith, Shavon Jackson-Michel, Timothy Fairchild

**Affiliations:** 1Clinical Research Australia, Perth, WA 6023, Australia; steve@clinicalresearch.com.au; 2College of Science, Health, Engineering and Education, Murdoch University, Perth, WA 6150, Australia; t.fairchild@murdoch.edu.au; 3DolCas Biotech, LLC, Landing, NJ 07850, USA; sjmichel@dolcas-biotech.com

**Keywords:** osteoarthritis, knee, curcumin, turmeric, clinical trial

## Abstract

Curcumin, a phytochemical from the spice turmeric, has anti-inflammatory properties and has been shown to have pain-relieving effects. In this 8-week, randomised, double-blind, placebo-controlled study, 101 adults with knee osteoarthritis received either 500 mg twice daily of a standardised curcumin extract (Curcugen^®^) or placebo. Outcome measures included the Knee Injury and Osteoarthritis Outcome Score (KOOS), knee pain ratings, Japanese Orthopaedic Association Score for Osteoarthritic Knees (JOA), PROMIS–29, and performance-based testing comprising the 40-m fast-paced walk test, 6-min walk test, timed up-and-go test, and 30-s chair stand test. Compared to the placebo, curcumin significantly reduced the KOOS knee pain score (*p* = 0.009) and numeric knee pain ratings (*p* = 0.001). Curcumin was also associated with greater improvements (*p* ≤ 0.05) than the placebo on the timed up-and-go test, 6-min walk test, and the JOA total score; but not the 30-s chair stand test or 40-m fast-paced walk test. Pain-relieving medication was reduced in 37% of participants on curcumin compared to 13% on placebo. The findings support the potential efficacy of curcumin for the treatment of osteoarthritis of the knee but studies of longer duration, varying treatment doses, differing curcumin extracts, and the use of other objective outcome measures will be helpful to expand on these findings.

## 1. Introduction

Osteoarthritis (OA) is a disease of the whole joint comprising structural alterations in the articular cartilage, subchondral bone, ligaments, capsule, synovial membrane, and periarticular muscles [1,2]. Knee OA accounts for almost 80 percent of the burden of OA worldwide and the prevalence of knee OA increases with obesity and age [3]. In 2020, knee OA was estimated to affect approximately 650 million people globally [4]. Management strategies for knee OA comprise non-pharmacological treatments such as education, exercise, and weight loss; while pharmacological treatments typically include paracetamol and nonsteroidal anti-inflammatory drugs (NSAIDs) [2,5]. Historically, paracetamol was recommended as the first-line medication for OA; however, in a 2017 meta-analysis, it was concluded that the effects of paracetamol compared to placebo were minimal [6]. Oral NSAIDs have generally demonstrated greater treatment efficacy, with moderate treatment effect sizes; however, there are safety concerns associated with their long-term use [6,7]. These include an increased risk of gastrointestinal bleeding, cardiovascular events, and NSAID-induced nephrotoxicity [8]. Other pharmacological options include opioids or intra-articular corticosteroids, hyaluronic acid, or triamcinolone. These treatments exhibit small to moderate treatment efficacy in reducing pain and improving function, but are often associated with adverse effects and typically, only short-term treatment efficacy [7].

Turmeric, the dried or fresh rhizome of the plant *Curcuma longa L*., is a common cooking spice that has also been extensively used in traditional medicine. Turmeric contains several phytochemicals such as turmerones and various polysaccharides; however, curcuminoids are believed to be responsible for most of turmeric’s therapeutic activity [9]. Curcuminoids are a mixture of curcumin, demethoxycurcumin, bisdemethoxycurcumin, and cyclocurcumin, with curcumin receiving the greatest attention as a natural treatment option for a range of inflammatory and pain-related conditions [10,11]. As a treatment for OA, there have been several trials that demonstrate it may reduce pain and increase function in OA sufferers. In two recent meta-analyses, it was confirmed that curcumin and turmeric extracts may be effective for the treatment of knee OA, with minimal adverse effects [12,13]. However, most of these studies have not included functional assessments and many have been evaluated as having a high risk of bias leading to some uncertainty about the true effect of curcumin for the treatment of knee OA [13]. The aims of this study were to examine the pain-relieving and functional effects of a curcumin extract (Curcugen^®^) using validated self-report and objective performance-based functional assessments. As most studies have been conducted in Asian countries, we were also interested in examining the efficacy and tolerability of curcumin in an Australian population that typically includes diverse ethnic groups.

## 2. Materials and Methods

### 2.1. Study Design

This was a two-arm, parallel-group, 8-week, single-centre, randomised, double-blind, placebo-controlled trial (Figure 1). All participants gave informed consent, and the trial protocol was approved by the Human Research Ethics Committee at the National Institute of Integrative Medicine (approval number 0072E_2020). This study was prospectively registered with the Australian and New Zealand Clinical Trials Registry (Trial ID. ACTRN12620000976987).

### 2.2. Sample Size Calculation

An a priori power analysis was undertaken to estimate the required sample size based on a single outcome variable [Knee Injury and Osteoarthritis Outcome Score (KOOS)—pain subscale score]. As there were no identified placebo-controlled studies using the KOOS in curcumin trials, we estimated the sample size based on studies using the Western Ontario and McMaster Universities Osteoarthritis Index (WOMAC) pain score as an outcome measure. In a meta-analysis comprising 5 studies, an effect size of 0.787 compared to the placebo was identified [14]. However, as higher doses were often administered, a more conservative effect size of 0.55 was predicted. Assuming a power of 80% and a type one error rate (alpha) of 5%, the number of participants required per group to find an effect on the KOOS pain subscale score was estimated as 42. We predicted a 15% dropout rate, which required us to recruit 50 participants per group to give us suitable power to find an effect compared to the placebo.

### 2.3. Recruitment and Randomisation

Participants were recruited through social media advertisements throughout the Perth (Western Australia) metropolitan region between November 2020 and January 2021. Interested volunteers were directed to a website page that included details about the study and a link to complete an online screening form for self-reported knee pain; a previous diagnosis of OA of the knee; the effect of knee pain on daily activities; the duration and severity of knee problems; prescription medication use; previous and current treatments for knee pain; history of medical or psychiatric disorders; alcohol, nicotine, and other drug use; and supplement and vitamin intake. If considered likely eligible, volunteers participated in a telephone interview comprising a structured series of questions to further assess their eligibility and to obtain additional demographic details. Suitable participants were then required to complete an online consent form and attend a face-to-face assessment located at our offices approximately 3 to 10 days after the phone interview. During the in-person assessment, respondents completed electronic versions of the KOOS, Patient-Reported Outcomes Measurement Information System-29 (PROMIS-29) and knee pain rating over the last week, ranging from 0 (no pain) to 10 (worst pain possible). Participants also completed a performance-based assessment comprising the 30-s chair stand test, 40-m fast-paced walk test, timed up-and-go test, and 6-min walk test. A researcher also completed the Japanese Orthopaedic Association Score for Osteoarthritic Knees (JOA).

Eligible and consenting participants were randomly allocated in a 1:1 ratio into one of two groups (Curcumin extract or placebo). A randomisation calculator (http://www.randomization.com (accessed on 1^st^ October 2020)) was used to ensure sequence concealment, with the randomisation structure comprising 10 randomly permuted blocks, containing 10 participants per block. The participant identification number was assigned based on the order of participant enrolment in the study. All capsules were packed in identical containers labelled by two intervention codes (held by the capsule manufacturer until final data analysis). Participants and study investigators were blind to treatment group allocation until all outcome data were collected and analysed. Participants were compensated for their time and travel costs with a $30 gift voucher which was given at the end of each face-to-face assessment. At the end of the study, participants allocated to the placebo condition were also offered a free 8-week supply of curcumin capsules.

### 2.4. Participants

Inclusion criteria: Male and female participants aged 45 to 70 years, with a body mass index between 20 and 35, a knee pain rating of at least 6 over the previous week (0 = no pain and 10 = worst pain possible), and knee pain for 3 months or more were recruited for this trial. All participants were required to have been formally diagnosed with OA of at least one knee by a medical professional (general practitioner and/or orthopaedic doctor/surgeon). The diagnosis was required to be based on minimum, a physical examination, interview of presenting symptoms, and other intra-articular pathologies such as bone marrow oedema, meniscal lesions, or cruciate ligament damage were required to be ruled out as causes of clinical symptoms. Participants were required to meet the National Institute for Health and Care Excellence clinical criteria for a diagnosis of knee OA comprising (1) age 45 years or older, (2) has activity-related knee joint pain, and (3) has either no morning joint-related stiffness or morning stiffness that lasts no longer than 30 min. To be included in the study, participants also had no plan to commence new treatments over the study period, were fluent in English and consented (via an online consent form) to all pertinent aspects of the trial.

Exclusion criteria: Participants were ineligible to participate in the study if they experienced a rapid worsening of knee symptoms, had a hot swollen knee, had a previous knee replacement on the affected knee, had a history of significant trauma to the affected knee, had a joint injection or arthroscopy in the past 6 months, or had any major surgery in the last year. Participants were also excluded if they were currently taking, or in the past 3 months were taking, pain-relieving medications (e.g., nonsteroidal anti-inflammatory drugs, analgesics, and corticosteroids) greater than twice a week, used anticoagulants in the preceding 3 months, were receiving cortisone or hyaluronic acid injections (or expected to receive such treatments during the study period), were diagnosed with gout or pseudogout within the last 3 months, had a history of gout in the knee joint, were diagnosed with other inflammatory arthritides (e.g., rheumatoid arthritis), septic arthritis, and malignancy (bone pain), or were diagnosed with a complex pain disorder or severe immobility (e.g., complex regional pain syndrome, severe back pain, multiple sclerosis, muscular dystrophy, Parkinson’s disease, or hemiplegia). Participants with a recent diagnosis of, and/or currently suffering from, unmanaged medical conditions including but not limited to: diabetes, uncontrolled hypertension, cardiovascular disease, gallbladder disease/gallstones/biliary disease, endocrine disease, or malignancies were excluded from participating in the study. Participants were also ineligible for the study if they were diagnosed with a significant neurological disease (e.g., Alzheimer’s disease, intracranial haemorrhage, or head/brain injury), or psychiatric disorder (other than mild-to-moderate depression/anxiety). People unable to walk unaided (e.g., requiring the use of a frame or walking stick), were house-bound due to immobility, consumed greater than 14 standard alcoholic drinks per week, was currently using or 12-month history of illicit drug abuse, or were currently taking supplements that may affect the treatment outcome (e.g., chondroitin, glucosamine, or curcumin) were ineligible to participate in the study. Finally, pregnant women, women who were breastfeeding, or women who intended to fall pregnant were not recruited for this study.

### 2.5. Interventions

Curcumin and placebo capsules were identical in appearance, being matched for colour coating, shape, and size. The active treatment, supplied by DolCas Biotech, LLC., contained 500 mg of a standardised curcuminoids extract (Curcugen^®^) in each capsule. Curcugen^®^ is a dispersible, 98.5% turmeric-based ingredient, containing 50% curcuminoids, 1.5% essential oils, and other native turmeric molecules, including turmeric polysaccharides. The placebo capsules contained the same excipients as the active capsules (microcrystalline cellulose). All participants were instructed to take 1 capsule, twice daily, with food, for 8 weeks. Capsule adherence was assessed by asking participants to provide an estimate of the consistency of capsule intake every week and a count of returned capsules at week 8 was also conducted. Treatment blinding was assessed by asking participants to predict group allocation (placebo, curcumin, or unsure) at the end of the study. All capsule bottles were provided to participants with instructions for use provided on the bottles. An information sheet about curcumin and what to do if a dose was missed was also given to participants. This information was also verbally conveyed to participants during their initial in-person assessment.

### 2.6. Outcome Measures

#### 2.6.1. Primary Outcome Measure:

Knee Injury and Osteoarthritis Outcome Score (KOOS): The KOOS is an extension of the WOMAC Osteoarthritis Index and is a validated self-report questionnaire designed to assess short and long-term patient-relevant outcomes associated with knee injury and pain [15,16]. It consists of 42 items where the following subscale scores can be calculated: (1) Knee symptoms and stiffness; (2) Knee pain; (3) Function in daily living; (4) Function in sports and recreational activities; and (5) Quality of life. A normalised score is determined by calculating the mean score in each subscale, dividing by 4 (each item within the subscale is scored on a scale from 0 to 4), multiplying this by 100, and subtracting this score from 100. The primary outcome measure for this study comprised the knee pain subscale score, with the remaining subscale scores used as secondary outcome measures. The KOOS was completed at baseline, week 4, and week 8.

#### 2.6.2. Secondary Outcome Measures:

Japanese Orthopaedic Association Score for Osteoarthritic Knees (JOA): The JOA is a reliable and valid observer-based knee scoring tool for evaluating functional status in patients with knee OA [17]. It consists of four domains with a maximum of 100 points: pain on walking (30 points), pain on ascending or descending stairs (25 points), mobility (35 points), and joint effusion (10 points). The JOA was completed by a researcher during the in-person assessments with participants at baseline and week 8.

Patient-Reported Outcomes Measurement Information System-29 (PROMIS-29): The PROMIS-29 is a validated, generic health-related quality of life self-report survey that assesses the following seven domains: (1) Physical function, (2) Anxiety, (3) Depression, (4) Fatigue, (5) Sleep disturbance, (6) The ability to participate in social roles and activities, and (7) Pain interference and intensity [18]. Higher scores on physical function score and the ability to participate in social roles and activities score indicate better function, whereas, lower scores in the other domains represent an improvement in symptoms. The PROMIS-29 was completed at baseline, week 4, and week 8.

Numeric knee-pain rating. Numeric knee pain ratings are a commonly-used and validated measure of acute pain and pain associated with OA of the knee [19,20]. Respondents rated the severity of their pain over the previous week on an 11-point scale ranging from 0 (no pain) to 10 (worst pain possible). Knee pain ratings were completed online every week.

Performance testing. Performance-based tests comprising the 40-m fast-paced walk test, 6-min walk test, timed up-and-go test, and 30-s chair stand test were completed at baseline and week 8. These performance-based measures assess physical function in adults with OA of the knee and are recommended as validated outcome measures by the Osteoarthritis Research Society International (ORSI) [21,22]. Based on the protocol detailed by the OSRI steering committee [23], each task comprised the following:30-s chair stand test—from a sitting position, participants stood up completely so their hips and knees were fully extended, then completely back down, so that their bottom fully touched the seat. The chair had no arms, and from the seated position the chair height was 50 cm. The number of repetitions over 30 s was recorded.40-m fast-paced walk test—the amount of time taken for participants to walk 4 by 10-m lengths where they were required to change direction 180 degrees and walk around a cone after every 10-m length was recorded.6-min walk test—participants walked as quickly as possible for six minutes to cover as much ground as possible. The distance travelled was recorded. The distance between cones was 17 m.Timed up-and-go test—participants were asked to stand up, walk around a cone 3 m away, and return to sit back in the chair at their regular pace. The chair had no arms, and from the seated position the chair height was 50 cm. The time to complete this task was recorded.

Analgesic pain medication use: via an online questionnaire, participants were asked to record the type, number, and dose of pain-relieving medications taken over the last 7 days.

Adverse events: Tolerability and safety of capsule intake by participants were assessed every week via an online open-ended question querying adverse effects believed to be associated with capsule intake. Participants were also requested to contact the researchers immediately if any adverse effects were experienced.

### 2.7. Statistical Analysis

For baseline data, an independent samples *t*-test was used to compare group data for continuous variables, and a Pearson’s Chi-square test was used to compare categorical data. The normality of residuals was assessed by the visual inspection of Q-Q plots and analysis of skewness and kurtosis. This demonstrated that data from the KOOS, JOA, knee pain ratings and most PROMIS-29 subscale scores were normally distributed, making parametric tests an appropriate option. However, results from the performance-based test were not normally distributed and transformations could not improve normality. As a result, non-parametric tests (Wilcoxon signed-rank test and Mann-Whitney U test) were used to analyse the data from these outcome measures.

KOOS subscale scores (weeks 0, 4, and 8), knee pain ratings (baseline, mean of weeks 1 to 4, mean of weeks 5 to 8), PROMIS-29 subscale scores (weeks 0, 4, and 8), and JOA (baseline and week 8) scores were analysed by using a repeated-measures, generalised mixed-effects model with adjustments for age, sex, BMI, medication change, and corresponding baseline values. To assess the clinical significance of change, between-group differences in the number of participants achieving a minimal clinical important difference (MCID) on the KOOS sub-scale scores were examined using a Pearson’s Chi-square test. MCID values were obtained from Jacquet et al. [24]. To examine between-group differences in the use of pain-relieving medications, the frequency of change in pain-relieving medication (increase, decrease, no change) from baseline to week 8 was examined using a Pearson Chi-Square test. All data were analysed using SPSS (version 26; IBM, Armonk, NY, USA). The critical *p*-value was set at *p* ≤ 0.05 for all analyses.

## 3. Results

### 3.1. Study Population

As detailed in Figure 1, from 214 people who completed the initial online screening questionnaire, 103 individuals did not meet the eligibility criteria, and 10 individuals withdrew consent to participate in the study. There were 101 volunteers who participated in the study with 96 people completing the study. Details of participant demographic information and baseline scores of the total recruited sample are detailed in Table 1. Demographic details and outcome scores were equivalent in both groups at baseline. All participants reported having an X-ray to confirm OA diagnosis and 71 reported to have additionally undergone a magnetic resonance imaging (MRI) of the knee. However, more details of OA classification and severity based on these diagnostic measures was not obtained from participants. Five participants withdrew from the study comprising 3 people who gave no reason for withdrawal, one person who was required to travel for work, and one person in the placebo group who reported experiencing ongoing stomach pain.

### 3.2. Outcome Measures

#### 3.2.1. Primary Outcome Measure: KOOS Knee Pain Subscale Score

As demonstrated in Table 2 and Figure 2, over the 8 weeks, KOOS knee pain subscale scores improved more in the curcumin group [11.98 (95% CI, 7.38 to 16.59)] compared to the placebo group [5.52 (95% CI, 0.75 to 10.28)] (*p* = 0.009) with a Cohen’s d effect size of 0.39. As detailed in Table 3, an examination of between-group differences in the number of participants achieving a MCID in the KOOS pain score revealed there were more participants having a MCID in the curcumin group (33%) compared to the placebo group (19%) (*p* = 0.047).

#### 3.2.2. Secondary Outcome Measures: Remaining KOOS Subscale Scores

As demonstrated in Table 2, there were no statistically-significant between-group differences in change in the remaining KOOS subscale scores. Moreover, there were no between-group differences in the number of participants achieving a MCID in these remaining KOOS subscale scores.

#### 3.2.3. Secondary Outcome Measure: Knee Pain Rating

As demonstrated in Table 2, over the 8 weeks, knee pain ratings improved more in the curcumin group [−2.09 (95% CI, −1.61 to −2.58)] compared to the placebo group [−1.57 (95% CI, −1.07 to −2.08)] (*p* = 0.001) with a Cohen’s d effect size of 0.30.

#### 3.2.4. Secondary Outcome Measure: JOA

Over the 8 weeks, the total JOA score improved more in the curcumin group [9.59 (95% CI, 6.34 to 12.84)] compared to the placebo group [1.11 (95% CI, −2.25 to 4.47)] (*p* < 0.001) with a Cohen’s d effect size of 0.74.

#### 3.2.5. Secondary Outcome Measure: PROMIS-29 Subscale Scores

Changes in the PROMIS-29 subscale scores across the two treatment groups are detailed in Table 2. There were no statistically-significant between-group differences in change in PROMIS-29 scores over time.

#### 3.2.6. Secondary Outcome Measure: Performance Testing Scores

Changes in scores on the performance-based tests are detailed in Table 4. At baseline, results from 5 participants were not obtained on the 30-s chair stand test as they refused to complete the task. A Wilcoxon Signed Rank Test revealed that in the curcumin group, there were statistically-significant improvements in the 30-s chair stand test (*p* < 0.001), timed up-and-go test (*p* < 0.001), and 6-min walk test (*p* = 0.005), but not the 40-m fast-paced walk test. In the placebo group, there was a statistically-significant improvement in the 30-s chair stand test (*p* < 0.001) only. A Mann-Whitney U test revealed there were statistically-significant greater improvements in the curcumin group compared to the placebo group in the timed up-and-go test (*p* = 0.032) and the 6-min walk test (*p* < 0.001), with effect sizes compared to the placebo of 0.38 and 0.63, respectively.

#### 3.2.7. Pain Medication Use

As detailed in Table 1, 49% and 53% of participants in the placebo and curcumin group, respectively, reported taking at least one pain-relieving medication the week before their initial assessment. Compared to baseline, at week 8 there was a statistically significant between-group difference in change in medication (*p* = 0.015). As detailed in Table 5, pain-relieving medication use decreased in 37% (19 out of 51) and 13% (6 out of 47) of participants in the curcumin and placebo group, respectively and increased in 11% (5 out of 47) and 21% (10 out of 47) of participants in the curcumin and placebo group, respectively.

#### 3.2.8. Intake of Supplements

Every week participants reported (as a percentage) the consistency of their capsule intake, and at week 8 returned capsule bottles with remaining capsules. Based on these details, only one participant (out of the 96 people who completed the study) did not take greater than 80% of their capsules.

#### 3.2.9. Efficacy of Participant Blinding

To evaluate the efficacy of condition concealment throughout the study, participants were asked at the end of the study to predict condition allocation (i.e., placebo, curcumin, or unsure). The efficacy of group concealment was high as 70% of participants either incorrectly guessed treatment allocation or were unsure.

#### 3.2.10. Adverse Events

The frequency of self-reported adverse events is detailed in Table 6. There was a tendency for more reported adverse events in the placebo group (*n* = 10) compared to the curcumin group (*n* = 5). No serious adverse events were reported by participants, although one participant in the placebo group withdrew due to reported stomach pain. There were no reports of any adverse events in 88% of participants in the curcumin group and 84% of participants in the placebo group. There were no statistically-significant between-group differences in changes in weight (*p* = 0.171), or systolic (*p* = 0.772) and diastolic (*p* = 0.900) blood pressure readings over time.

## 4. Discussion

In this randomised, double-blind, placebo-controlled study, the effects of 8 weeks of supplementation with a curcumin extract at a dose of 500 mg twice daily on adults with knee OA was examined. Compared to the placebo, curcumin significantly improved the KOOS knee pain score (primary outcome measure) and numeric knee pain ratings. These improvements in the KOOS knee pain score and knee pain ratings had medium effect sizes of 0.39 and 0.30 compared to the placebo, respectively. The MCID represents the smallest change in a treatment outcome that an individual would identify as important, and based on MCID scores identified by Jacquet et al. [24], 39% of participants reported a MCID in the KOOS pain score compared to 19% in the placebo group. Moreover, according to Roos and Lohmander [25], an increase of 10 points or more on KOOS subscale scores represent a clinically-significant difference. In this study, curcumin administration was associated with an approximate 12-point improvement in the KOOS knee pain score after the 8-week intervention, indicating curcumin had both a clinically and statistically-significant improvement in pain reporting. However, curcumin was not associated with improvements in the other KOOS subscale scores comprising knee symptoms, function in daily living, function in sport and recreation, and knee-related quality of life. Based on the results of the PROMIS-29, as a measure of overall quality of life, there were also no significant between-group differences in change in scores over time.

To support findings from the self-report measures, an observer-based measure (JOA) and several performance-based assessments were conducted. Compared to the placebo, curcumin was associated with greater improvements in the JOA total score, timed up-and-go test, and the 6-min walk test, but not the 30-s chair stand test or 40-m fast-paced walk test. The JOA is an observer-based rating system that uses patient self-report and physical observations to evaluate four domains comprising pain on walking, pain on ascending or descending stairs, range of motion, and joint effusion [17]. The baseline timed up-and-go test results (mean: 6.65 s) for the cohort in the present study were faster than those previously reported in a similarly-aged knee-OA cohort (10.9 ± 3.6 s) [26] and comparable to those in a ‘healthy’ Australian cohort (7.4 ± 1.9 s) [27]. Baseline 6-min walk test results (518.5 m) are lower than those reported in a similarly-aged ‘healthy’ Australian cohort (659 ± 62 m) [28], but higher than those reported previously in knee-OA patients (416.4 m) [29]. These results suggest that curcumin intake was associated with objective functional improvements in adults with knee OA, even in patients maintaining relatively high levels of physical function.

Curcumin was well tolerated, there were no significant adverse effects reported, and the frequency of adverse effects was similar in both the curcumin and placebo groups. Curcumin supplementation was also associated with a greater use of pain-relieving medication compared to the placebo. From baseline to week 8, 37% of participants in the curcumin group reported decreasing their pain-relieving medication, compared to only 13% of participants in the placebo group. Moreover, 11% of participants in the curcumin group reported increasing their pain-relieving medication, compared to 21% in the placebo group.

Several of the findings in this study are consistent with previous trials examining the efficacy of curcumin for knee OA. In a recent meta-analysis, it was confirmed based on findings from 16 randomised controlled trials and 1810 adults, curcumin/turmeric extracts significantly reduced self-reported measures of knee pain (effect size = −0.82) and improved self-reported physical function (effect size = −0.75) compared to the placebo, and had similar treatment efficacy as NSAIDs [13]. We observed similar reductions in pain ratings, albeit with a lower effect size, but no change in self-report measures of physical function. It is important to highlight that the non-significant between-group differences in the KOOS subscale scores representing function in sport and recreation and knee-related quality of life could be partly attributed to the high placebo effect, as there were mean improvements of approximately 10 points in the two subscales (compared to 5-point changes in the other subscale scores). Interestingly, in this meta-analysis, most studies were conducted in Asia (e.g., India, Thailand, and Iran) while only two studies were conducted in non-Asian countries (Belgium and Australia). Randomised controlled trials conducted in Asia were associated with significantly-larger treatment effects compared to those conducted in other countries. Wang et al. [13] concluded that lower BMI, and to a lesser extent, younger age, are associated with greater treatment efficacy from curcumin. However, the age and BMI of participants in this study were similar to the previously conducted trials. Baseline severity in knee OA could account for the differences in treatment efficacy and require further investigation. It is also possible that the pain-relieving effects of curcumin may vary across ethnic groups with a tendency for reduced efficacy in non-Asians. Therefore, generalising the results from studies conducted on different ethnic groups should be done with caution [13]. In a recent well-designed randomised trial conducted in Australia, effect sizes from curcumin supplementation in adults with knee OA were lower than others reported [13], which further suggests ethnic-based differences in efficacy [30]. However, in contrast to the study by Wang et al. [30], where they used the WOMAC as an outcome measure, we only identified significant between-group differences in the KOOS pain subscale score. Improvements in the performance-based assessments comprising a 6-min walk test and timed up-and-go test were identified in our study, while no differences were observed in the 30-s chair stand-test or 40-m fast-paced walk test. These results partly support those of Wang and colleagues [30], who found no improvements in the 30-s chair-stand test or the 40-m fast-paced walk test. However, the 6-min walk test and timed up-and-go test were not part of the functional assessment battery adopted by Wang and colleagues.

In comparison to pharmaceutical drug-based trials, the treatment effects from curcumin administration in this trial are at least as comparable. In a network meta-analysis comparing the efficacy of different pain-relieving medications for the treatment of pain in adults with knee and hip OA, effect sizes ranged from −0.63 to −0.07, compared with placebo [6]. Paracetamol was associated with the lowest treatment effect (−0.07 to −0.18 for doses ranging from less than 2000 mg to 4000 mg), ibuprofen had medium treatment effects (−0.30 to −0.42 for doses of 1200 mg and 2400 mg, respectively), while rofecoxib at a dose of 50 mg daily was associated with the largest treatment effect of −0.63. Topical NSAIDs have been shown to have an effect size of −0.30 compared to placebo for the treatment of OA-related pain [31]. The effect sizes of −0.39 (KOOS knee pain score) and −0.30 (knee pain ratings) identified in this study, and the strong safety profile, suggests that a curcumin extract (Curcugen^®^) delivered at a dose of 1000 mg daily for 8 weeks is a promising, well-tolerated, non-pharmaceutical, treatment option. While the strong safety profile in the current study accords with previous research [13,30], the safety and efficacy associated with its long-term use require investigation in future trials. In previous trials on curcumin, doses of up to 2000 mg have been investigated [32] and the longest treatment duration was 16 weeks [33].

While the underlying mechanisms explaining the therapeutic benefits of curcumin are incompletely understood, the anti-inflammatory and antioxidant effects of curcumin are believed to be associated with its therapeutic effects in OA. The anti-inflammatory properties of curcumin are similar to NSAIDs as it can influence nuclear factor (NF)-kB activity and reduce concentrations of pro-inflammatory cytokines such as interleukins, phospholipase A2, and 5-lipoxygenase (5-LOX), and lower cyclooxygenase-2 (COX-2) activity [34,35]. Articular cartilage damage and chondrocyte apoptosis are common features in OA, and in animal and in vitro models curcumin exhibited protective effects on degeneration in articular cartilage. Curcumin suppressed interleukin (IL)-1β-induced chondrocyte apoptosis by activating autophagy and restraining NF-κB signalling pathway [36]. X-ray and histopathological images have also revealed curcumin can restore joint architecture and reduce joint swelling induced by monosodium iodoacetate-induced knee OA in rats. Curcumin treatment also lowered concentrations of inflammatory mediators such as tumour necrosis factor (TNF)-α, IL-1β, IL-6, and C-reactive protein (CRP); the expression of matrix metalloproteinase-3, 5-LOX, COX-2, and NF-κB in synovial tissue; and reduced oxidative stress and increased antioxidant enzyme activity [37]. In human trials, the anti-inflammatory and antioxidant effects of curcumin supplementation in adults with knee OA are inconsistent. In a 3-month trial, compared to placebo, curcumin supplementation did not affect the erythrocyte sedimentation rate (ESR) but reduced concentrations of CRP, and the percentage of CD4+ and CD8+ T cells, Th17 cells and B cells [38]. In a comparative study with the NSAID, diclofenac, 4 weeks of curcumin supplementation lowered the secretion of COX-2 by monocytes in synovial fluid at similar levels to diclofenac [39]. In another comparative drug trial, 8-weeks of curcumin supplementation reduced concentrations of CRP and TNF-α, whereas paracetamol administration did not affect these markers [40]. However, in another OA study, compared to the placebo, 6 weeks of curcumin supplementation did not significantly alter the ESR or concentrations of IL-4, IL-6, CRP, TNF-α [41]. Investigations into the antioxidant effects of curcumin in people with OA have revealed that 6 weeks of curcumin supplementation increased serum superoxide dismutase and glutathione activity; and lowered concentrations of malondialdehyde compared with the placebo [42]. In another placebo-controlled trial, concentrations of reactive oxygen species and malondialdehyde were significantly lower compared to the placebo after 120 days of curcumin supplementation [43]. Examinations into the effects of curcumin on cartilage degradation revealed that 3 months of curcumin supplementation lowered levels of Coll2-1, a biomarker of cartilage degradation, but this was not significantly different to the placebo [44]. In a 12-week trial, despite having positive effects on knee pain, curcumin supplementation did not affect knee effusion-synovitis or cartilage composition [30]. The inconsistency in the biological effects of curcumin in these human trials highlights the need for further investigations to clarify the mechanisms associated with curcumin’s pain-relieving and functional effects in OA.

### Study Limitations and Directions for Future Research

Despite overall positive findings from this study, several factors influence the robustness and generalisability of the findings. People recruited for the trial reported having a diagnosis of knee OA by a medical professional (general practitioner and/or orthopaedic doctor/surgeon) which was confirmed by an X-ray and/or MRI, had symptoms strongly suggestive of OA of the knee, had other intra-articular pathologies such as bone marrow oedema, meniscal lesions, or cruciate ligament damage ruled out as causes of clinical symptoms, and described no other contributory causes of their pain; however, no formal clinical examination, ultrasonography, knee radiography, or magnetic resonance imaging was conducted by our research team. A more comprehensive assessment will be important in future trials to confirm knee OA, its severity, and other potential contributory causes. This may help to identify and classify individuals based on disease severity, symptomatic profile, and imaging pathology, and in turn, identify individuals that will achieve the greatest benefit from curcumin supplementation. In particular, classifications based on the Kellgren Lawrence (KL) system will help to identify OA severity and KL grade that will benefit from curcumin supplementation [45]. KOOS scores have been shown to be correlated with KL classifications, with KOOS symptom and pain scores below 50 suggestive of a grade 4 KL classification [46]. The mean baseline KOOS pain and symptom scores in people recruited in this trial was approximately 60 and only 10 percent of participants had a KOOS score below 50. This suggests that most people recruited in this study had a grade 3 KL classification or lower. Therefore, generalising the results from this study to people with severe knee OA pathology such as a grade 4 KL classification or in people with total knee arthroplasty should be avoided until further curcumin trials are conducted using KL classifications. Approximately 50 percent of recruited individuals reported taking pain-relieving medication at baseline, albeit at a frequency 2 times a week or less. The efficacy of curcumin supplementation as a stand-alone intervention could not, therefore, be adequately deduced from this trial. Further well-controlled investigations into the efficacy of curcumin supplementation as a stand-alone intervention or as an adjunct to different pain-relieving medications (or lifestyle interventions such as exercise) will be important to investigate in the future. In addition, despite some evidence in this trial suggesting curcumin supplementation was associated with the reduced use of pain-relieving medication, more comprehensive monitoring of medication frequency, type, and dose will be important. Meta-analyses have demonstrated that curcumin can be effective for the treatment of knee OA, however, further research is required to investigate the safety and efficacy of different doses, curcumin extracts, and treatment periods. To date, the longest curcumin trial was 4 months, and the highest administered dose of a curcumin extract was 2000 mg a day. Even though curcumin seems to have a strong safety profile, longer trials using different doses are required. Moreover, as this trial was only 8 weeks, longer trials will provide more definitive conclusions about the medium and long-term efficacy of curcumin supplementation. Understanding the safety and efficacy profiles of different curcumin extracts will also be important as curcumin is associated with poor bioavailability and the differing compositions of curcumin extracts may influence treatment outcomes and tolerability. As has already been discussed, treatment effect sizes differ substantially across curcumin trials on knee OA, with very large effect sizes identified in Asian-conducted studies. Understanding the reasons for this variability in effect sizes will be important in future trials. Recruitment strategies could be a possible explanatory cause; however, it is plausible that efficacy in non-Asian populations may be impacted by genetic factors, dietary habits, or even ethnic differences in the tolerability and absorption of curcumin. Moreover, variances in intestinal microbiota may also account for the differing treatment effects as the microbiome can influence the beneficial or deleterious effects of polyphenols [47,48]. As most curcumin trials have been conducted in Asian countries, further investigations across a diverse array of ethnic groups will be helpful to understand the beneficial effects and tolerability of curcumin. Finally, even though a robust double-blind, placebo-controlled trial was used in this trial, to further substantiate the efficacy of curcumin for the treatment of knee OA, future trials incorporating 3-arms (i.e., placebo, curcumin, and oral NSAID) will be important to compare the effects of curcumin with commonly-used, validated treatments.

## 5. Conclusions

In conclusion, in this randomised, double-blind, placebo-controlled study, 8 weeks of supplementation with a curcumin extract (Curcugen^®^) at a dose of 500 mg twice daily on adults with knee OA was associated with statistically-significant greater improvements in knee pain, the JOA total score, and performance-based tests (timed up-and-go test and 6-min walk test) compared to the placebo. However, compared to the placebo, curcumin supplementation did not significantly improve self-report measures assessing knee symptoms and stiffness, function in daily living, function in sports and recreational activities, and quality of life impacted by knee pain. Curcumin was well tolerated and there were findings to suggest reduced use of pain-relieving medications, although this latter finding requires further confirmation in trials with more stringent medication monitoring.

## Figures and Tables

**Figure 1 nutrients-14-00041-f001:**
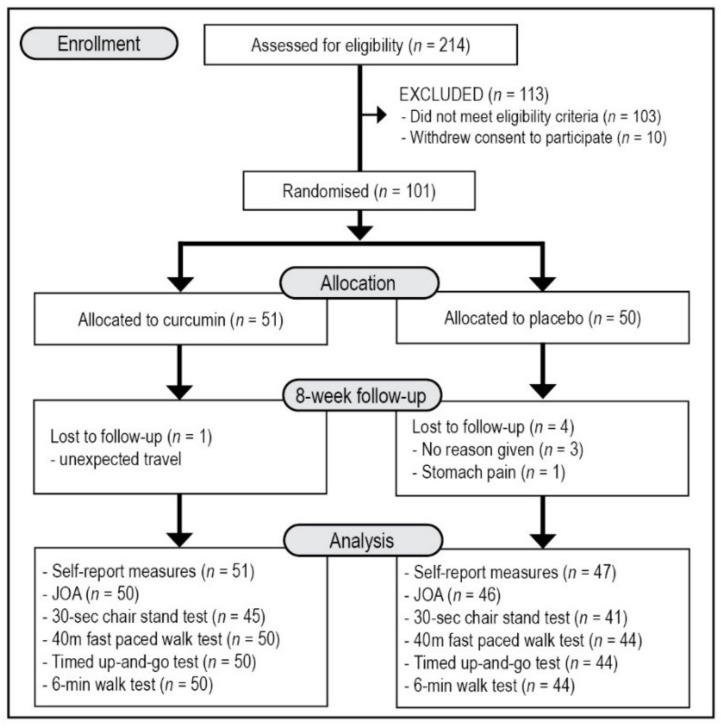
Systematic Illustration of Study Design. JOA = Japanese Orthopaedic Association Score for Osteoarthritic Knees, KOOS = Knee Injury and Osteoarthritis Outcome Score, PROMIS-29 = Patient-Reported Outcomes Measurement Information System-29.

**Figure 2 nutrients-14-00041-f002:**
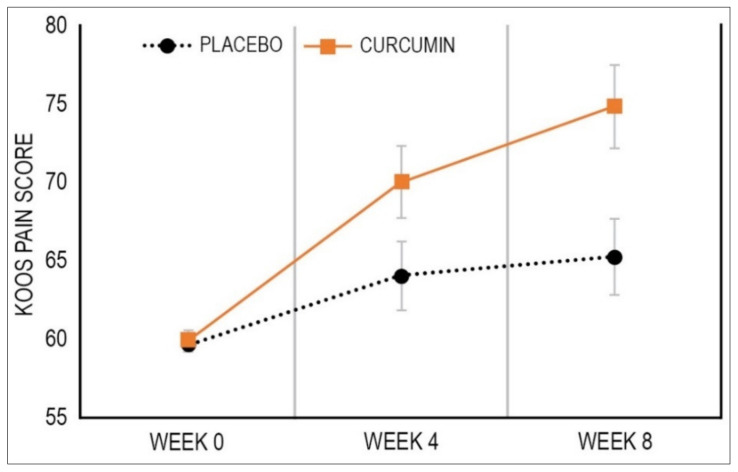
Mean Change in KOOS Pain Score. Data are estimates from linear mixed-effects model. Bars depict standard error.

**Table 1 nutrients-14-00041-t001:** Baseline Demographics Details and Questionnaire Scores.

		Placebo (*n* = 50)	Curcumin (*n* = 51)	*p*-Value
Age	Mean	57.92	59.59	0.195 ^a^
	SE	0.88	0.92	
Gender	Female (*n*)	26	24	0.619 ^b^
	Male (*n*)	24	27	
BMI	Mean	28.82	28.93	0.898 ^a^
	SE	0.60	0.65	
Systolic blood pressure (mmHg)	Mean	141.50	138.16	0.312 ^a^
	SE	2.54	2.09	
Diastolic blood pressure (mmHg)	Mean	87.56	87.08	0.793 ^a^
	SE	1.46	1.10	
Marital status	Single	9	11	0.653 ^b^
	Married/defacto	41	40	
Took pain-relieving medications during previous week	Yes (%)	49	53	0.692 ^b^
Educational level	Secondary	28	23	0.149 ^b^
	Tertiary	16	14	
	Post-graduate	6	14	
Exercise level	Never/rarely	14	11	0.629 ^b^
	1 to 2 times a week	4	6	
	3 to 5 times a week	16	13	
	6+ times a week	16	21	
KOOS—Pain	Mean	61.17	60.68	0.848 ^a^
	SE	1.93	1.68	
KOOS—Symptoms	Mean	59.29	59.00	0.922 ^a^
	SE	2.07	2.09	
KOOS—Daily Living	Mean	69.41	68.96	0.886 ^a^
	SE	2.39	2.08	
KOOS—Sports and recreational activities	Mean	41.00	43.77	0.559 ^a^
	SE	3.36	3.33	
KOOS—Quality of life	Mean	38.25	40.81	0.485 ^a^
	SE	2.72	2.43	
Knee pain rating	Mean	5.80	6.22	0.174 ^a^
	SE	0.22	0.20	
JOA—Total Score	Mean	70.90	69.90	0.753 ^a^
	SE	2.23	2.24	
PROMIS-29—Physical function	Mean	3.92	3.93	0.979 ^a^
	SE	0.09	0.10	
PROMIS-29—Anxiety	Mean	1.46	1.43	0.802 ^a^
	SE	0.09	0.08	
PROMIS- 29—Depression	Mean	1.35	1.36	0.954 ^a^
	SE	0.08	0.09	
PROMIS-29—Fatigue	Mean	2.16	2.12	0.820 ^a^
	SE	0.13	0.12	
PROMIS-29—Sleep disturbance	Mean	2.52	2.66	0.476 ^a^
	SE	0.13	0.15	
PROMIS-29—Social roles	Mean	3.55	3.63	0.660 ^a^
	SE	0.14	0.13	
PROMIS-29—Pain interference	Mean	2.44	2.45	0.933 ^a^
	SE	0.11	0.14	
PROMIS-29—Pain intensity	Mean	4.28	4.69	0.245 ^a^
	SE	0.22	0.27	
30-s chair stand test	Mean	17.02	15.86	0.280 ^a^
	SE	0.75	0.77	
40 m fast paced walk test	Mean	25.87	26.18	0.708 ^a^
	SE	0.64	0.56	
Timed up-and-go test	Mean	6.46	6.84	0.126 ^a^
	SE	0.17	0.18	
6-min walk test	Mean	522.15	515.42	0.669 ^a^
	SE	12.61	9.40	

^a^ = independent samples *t*-test; ^b^ = chi-square test.

**Table 2 nutrients-14-00041-t002:** Change in Outcome Measures from Baseline to Week 8.

Endpoints	Placebo (*n* = 47)	Curcumin (*n* = 51)	Mean Between-Group Difference in Change (95% CI)	*p*-value ^†^	Cohen’s d
	Mean Change (95% CI)	Mean Change (95% CI)			
**Primary**					
KOOS—Pain (Primary)	5.52 (0.75 to 10.28)	11.98 (7.38 to 16.59)	6.47 (0.18 to 12.75)	0.009	0.39
**Secondary**					
Knee pain rating	−1.57 (−2.08 to −1.07)	−2.09 (−2.58 to −1.61)	−0.52 (−1.19 to 0.14)	0.001	0.30
KOOS—Symptoms	5.21 (0.45 to 9.98)	6.48 (1.86 to 11.09)	1.26 (−5.02 to 7.55)	0.473	0.08
KOOS—Daily living	5.42 (0.86 to 9.98)	8.20 (3.79 to 12.61)	2.78 (−3.23 to 8.79)	0.106	0.18
KOOS—Sports and recreation	10.13 (3.63 to 16.63)	8.79 (2.53 to 16.06)	−1.34 (−9.89 to 7.21)	0.454	0.06
KOOS—Quality of life	9.83 (3.83 to 15.84)	13.69 (7.92 to 19.47)	3.86 (−4.00 to 11.72)	0.624	0.19
PROMIS-29 Physical function	−0.01 (−2.11 to 0.19)	0.24 (0.05 to 0.43)	0.25 (−0.01 to 0.52)	0.082	0.36
PROMIS-28 Anxiety	0.07 (−0.10 to 0.24)	−0.14 (−0.03 to 0.03)	−0.21 (−0.43 to 0.01)	0.168	0.36
PROMIS-29 Depression	−0.10 (−0.25 to 0.04)	−0.14 (−0.28 to 0.00)	−0.04 (−0.23 to 0.15)	0.103	0.08
PROMIS-29 Fatigue	−0.20 (−0.40 to 0.01)	−0.06 (−0.25 to 0.14)	0.14 (−0.13 to 0.41)	0.370	0.20
PROMIS-29 Sleep	−0.14 (−0.36 to 0.08)	−0.15 (−0.36 to 0.07)	0.01 (−0.30 to 0.29)	0.553	0.01
PROMIS-29 Social roles	0.15 (−0.11 to 0.41)	0.08 (−0.16 to 0.33)	−0.07 (−0.41 to 0.27)	0.501	0.08
PROMIS-29 Pain interference	−0.45 (−0.68 to −0.23)	−0.47 (−0.68 to −0.25)	−0.01 (−0.31 to 0.28)	0.872	0.03
PROMIS-29 Pain intensity	−0.74 (−1.37 to −0.11)	−0.82 (−1.43 to −0.20)	−0.08 (−0.92 to 0.77)	0.550	0.04
JOA	1.11 (−2.25 to 4.47)	9.59 (6.34 to 12.84)	8.48 (4.03 to 12.92)	<0.001	0.74

Results are generated from generalised mixed-effects models adjusted for age, sex, body mass index, and corresponding baseline values. **^†^** P-values are generated from repeated measures generalised mixed-effects models adjusted for age, sex, body mass index, and corresponding baseline values (time x group interaction). For KOOS scores increased scores signify improvements. For knee pain ratings, reduced scores signify improvements. For PROMIS scores reduced scores signify improvements except for physical function and social roles, where an increase signifies improvements. For JOA scores, an increase signifies an improvement in function.

**Table 3 nutrients-14-00041-t003:** Number of Participants Achieving Minimal Clinically Important Difference (Based on KOOS Scores).

		Placebo (*n* = 47)	Curcumin (*n* = 51)	*p*-Value
KOOS—Symptoms	Yes	11	14	0.646
	No	36	37	
KOOS—Pain	Yes	9	20	0.045
	No	38	31	
KOOS—Daily living	Yes	11	10	0.647
	No	36	41	
KOOS—Sports and recreation	Yes	18	18	0.758
	No	29	33	
KOOS—Quality of life	Yes	13	17	0.543
	No	34	34	

**Table 4 nutrients-14-00041-t004:** Change in Performance-based Test Results.

		Placebo			Curcumin			Between-Group, *p*-Value ^b^
		Week 0	Week 8	Within-Group, *p*-Value ^a^	Week 0	Week 8	Within-Group, *p*-Value ^a^	
30-s chair stand test (repetitions)	n	41		<0.001 ^a^	45		<0.001	0.391
	Mean	17.32	20.54		16.29	19.91		
	SE	0.83	0.87		0.8	0.88		
40 m fast paced walk test (s)	n	44		0.795 ^a^	50		0.699	0.604
	Mean	25.6	25.68		26.28	26.11		
	SE	0.64	0.6		0.57	0.61		
Timed up-and-go test (s)	n	44		0.061 ^a^	50		<0.001	0.032
	Mean	6.43	6.22		6.86	6.36		
	SE	0.18	0.18		0.19	0.18		
6-min walk test (m)	n	44		0.480 ^a^	50		0.005	0.013
	Mean	525.81	521.07		512.3	530.18		
	SE	13.91	13.54		9.05	11.21		

^a^ = Wilcoxon Signed Rank Test; ^b^ = Mann-Whitney U test.

**Table 5 nutrients-14-00041-t005:** Change in pain-relieving medication at week 8 compared to baseline.

	Placebo (*n* = 47)	Curcumin (*n* = 51)	*p*-Value *
No Change	31	27	0.014
Decreased	6	19	
Increased	10	5	

* Person Chi-Square Test.

**Table 6 nutrients-14-00041-t006:** Frequency of self-reported adverse effects.

	Placebo	Curcumin
Loose stools/diarrhoea	1	3
Heartburn/indigestion		1
Stomach pain	2	1
Nausea	1	
Constipation	1	
Tiredness	2	
Irritability	1	
Mouth ulcer	1	
Pins and needles in foot	1	
Total number of adverse effects	10	5

## Data Availability

Data presented in this study are available on request from the corresponding author.

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
