# Peer review of "An Investigation into the Effects of a Curcumin Extract (Curcugen®) on Osteoarthritis Pain of the Knee: A Randomised, Double-Blind, Placebo-Controlled Study"

_nutrients, 2021, doi:10.3390/nu14010041_

Round 1

Reviewer 1 Report

This randomized trial compares curcumin or placebo in patients suffering from knee OA.

Despite the interesting purpose and precise design, there is a fatal flaw:

Participant were selected mainly considering subjective criteria.

Why authors did not include radiographic classification of OA and MRI findings?

There are several intra-articular pathologies affecting clinical symptoms and a broad spectrum of joint changes (i.e. bone marrow edema, meniscal lesions or ACL injuries) that should be properly assessed, classified and compared to demonstrate homogeneous study and control groups.

Moreover 8 weeks is a very short period to achieve strong conclusion, and longer follow-up is desirable to observe mid-term and long term efficacy of curcumin over placebo.

Reviewer 2 Report

Thank you for the possibility to review this interesting article. The main question in this article is curcumin influence on OA pain. The answer is described on the line 527 but can you explain no effect on the quality of life impacted by knee pain (531).

This article starts the very interesting topic according to future OA therapy possibilities or the novel drugs development but please consider to explain why gold standard three arm knee OA study was not implemented in this situation. Paper is well written with the clear figures. I think that not all NICE OA diagnosis criteria were described (125). Text is clear and easy to read even for the beginning scientist. There is lack information about hyaluronic acid and other iniections in exclusion criteria. The study question is answered but there are no limitations of the study clearly presented. Please explain the name”medical professional” who diagnosed OA.

Round 2

Reviewer 1 Report

Method section has been improved but data concerning MRI findings and X rays are not reported.

Authors should include data about X ray OA classification as diagnosis of osteoarthritis is too vague.

OA has a broad spectrum of articular changes from mild joint space narrowing to severe axial deformity, subchondral abnormalities and joint collapse. Authors should precisely classify the measured grade of OA possibly with Kellgren Lawrence classification (PMID: 13498604) and clearly report grade of included patients as is not believable that patients with severe OA with TKA indication may benefit from conservative measures.

The same for MRI findings of 71 patients who underwent MRI investigation. Authors should clearly report the detected intra-articular pathologies as there are benign signs and other intra-articular lesions with poor prognosis and all these aspect can significantly influence the measured outcome (selection bias).

In conclusion it is not clear if the populations are homogenous in terms of OA grade as a selection bias can significantly influence the final results.

Moreover authors should describe the ideal target for this specific therapy (i.e.: mild OA?). 

Finally the follow-up is very short and should be stated within the limitation section. 

Reviewer 2 Report

Manuscript can be accepted as the pilot study presenting short term observation.

Author Response

Thank you for your positive feedback